# Cell behavior on silica-hydroxyapatite coaxial composite

**Jesús Alberto Garibay-Alvarado[1], Ericka Berenice Herrera-Ríos[2], Claudia Lucía Vargas-Requena[1], Álvaro de Jesús Ruíz-Baltazar[3], Simón Yobanny Reyes-López[1]***

**1** Instituto de Ciencias Biomédicas, Universidad Autónoma de Ciudad Juárez, Ciudad Juárez, Chihuahua, México, **2** Departamento de Estudios de Posgrado e Investigación, Tecnológico Nacional de México campus Ciudad Juárez, Ciudad Juárez, Chihuahua, México, **3** CONACYT-Centro de Física Aplicada y Tecnología Avanzada, Universidad Nacional Autónoma de México, Querétaro, Querétaro, México

* simon.reyes@uacj.mx

## Abstract

Progress in the manufacture of scaffolds in tissue engineering lies in the successful combination of materials such as bioceramics having properties as porosity, biocompatibility, water retention, protein adsorption, mechanical strength and biomineralization. Hydroxyapatite (HA) is a ceramic material with lots of potential in tissue regeneration, however, its structural characteristics need to be improved for better performance. In this study, silica-hydroxyapatite ($SiO_2$-HA) non-woven ceramic electrospunned membranes were prepared through the sol-gel method. Infrared spectra, scanning electron microscopy and XRD confirmed the structure and composition of composite. The obtained $SiO_2$-HA polymeric fibers had approximately 230±20 nm in diameter and were then sintered at 800˚C average diameter decreased to 110±17 nm. Three configurations of the membranes were obtained and tested in vitro, showing that the composite of $SiO_2$-HA fibers showed a high percentage of viability on a fibroblast cell line. It is concluded that the fibers of $SiO_2$-HA set in a coaxial configuration may be helpful to develop materials for bone regeneration.

## 1. Introduction

Hydroxyapatite (HA) is a ceramic used in dental and medical applications due to its biocompatibility, bioactivity, and osteogenic characteristics [1], however, since HA is very brittle, its applications are limited. Synthetic hydroxyapatite (HA) has similar characteristics to those of the hydroxyapatite of bone, but poor biodegradation properties, which prevents natural bone growth for extended periods. Also, its low strength and fracture toughness have reduced the field of possible applications to only those where the implant will be subjected to low stress [2]. In designing HA-based materials to overcome various limitations, numerous research groups have proposed different interpretations on how to incorporate hydroxyapatite, as a coting in metal such as titanium showing high biocompatibility [3], as powder derived high-density composite ceramics [4] and as particles embedded in polymers [5,6], to name a few. Ideally, any attempt to enhance physical properties or to induce antibacterial effect should not compromise cellular functionality in terms of cell viability [7].

manuscript. All values and other measures are reported in the manuscript.

**Funding:** This research received no external and internal funding.

**Competing interests:** The authors have declared that no competing interests exist.

In an effort to counteract the fragility of HA, it has been mixed with alumina to form composite powders to be used as bone substitutes, showing good compatibility with human osteoblasts [4]. The antibacterial capability of HA has been tested as chemically modified particles embedded in a polymer composite, successfully reducing bacterial and fungal growth while improving the mechanical properties of said composite with HA concentrations as low as 2.5% [5]. The addition of HA, silver nanoparticles, corn silk extract, and hyaluronic acid to a β-TCP hydrogel greatly increases the antibacterial activity and also promotes differentiation on bone cells [6].

Silicon can have an influence over the mineralization process while also contributing to the proliferation of osseous cells [8] Silicon is an essential element for normal bone growth [9] and, since the discovery of bioglass, it has been used extensively for tissue regeneration [10]. Different attempts have been made in combining the properties of HA and silica, such as the incorporation of $SiO_2$ nanoparticles on HA slurry [8] and HA-based cement [11], the production of silicon-substituted hydroxyapatite materials [10], calcium phosphate glasses [12] and two-phase $SiO_2$-HA electrospun non-woven membranes [13]. This kind of ceramics has the capability of dissolving into silanol groups collecting over the surface of the material, which in turn will allow the formation of calcium phosphate; these ions can also stimulate the union between tissue and the material. Research on silica-gel has proven that it acts as an effective inducer of hydroxyapatite using the Ca and P available in the surrounding fluid [14].

Various techniques can be used for the fabrication of scaffolds, amongst them, electrospinning has allowed the fabrication of non-woven scaffolds with a large surface area from different compositions, capable of mimicking the matrix of tissues such as skin and bone [15–18]. The use of polymers and ceramics through the electrospinning technique can produce fibers with bioactive properties which can be potentially used in the dental and orthopedic fields for bone regeneration [17,18]. In order to know the effect of any given biomaterial, it is necessary a first step of testing the material before reaching humans. MTT assay has been performed on a variety of HA-based biomaterials to test their effect on the viability of cells [11,18–20], and since this approach is inexpensive, it always must be considered before animal testing. The aim of this work was to obtain $SiO_2$-HA non-woven ceramic membranes through electrospinning for the manufacture of bioceramics scaffolds. The cytotoxic behavior of the fibers was tested using the MTT colorimetric assay before performing an in-vivo study.

## 2. Materials and methods

### 2.1 Fibrillar composite obtainment

The fibrous composite was fabricated using a modified version of the methodology by Garibay-Alvarado *et al*. [13]. Hydroxyapatite sol was prepared using calcium nitrate tetrahydrate [$Ca(NO_3)_2 \cdot 4H_2O$] (Sigma-Aldrich®, 99%) dissolved in ethanol and triethyl phosphite ($C_2H_5O)_3P$ (Sigma-Aldrich®, 99%) was hydrolyzed in ethanol (Hycel®, 99.5%). The calcium nitrate solution was added by dripping to the triethyl phosphite during 1h. The solution was stirred vigorously for 24 h at 40°C and aged for 6 h at 60°C. The resulting sol was evaporated for 1 h to obtain a solid content of 80 w/v %. The silica precursor was prepared dissolving tetraethyl orthosilicate (TEOS) (Fluka®, 99%) in ethanol, followed by the adding of a solution of water and HCl used as catalyst under constant stirring for 30 min at room temperature, the proportions of the components were 1:2:2:0.1, respectively.

For the electrospinning process, a solution of PVP (1,000,000 M.w., Sigma-Aldrich®) in ethanol with concentration of 10 w/v % was prepared. This solution was mixed with the silica precursor at 10 w/v % and HA sol at 20 w/v %. The solutions were charged in syringes (Kendall® Monoject™) of 30 mL and electrospun with a Nabond® NEU-Pro™ device, alone, using a

feeding rate of 2 mL/h, 20 cm between needle and collector and a voltage of 15 kV; and together through a coaxial nozzle with double feeding, with a feeding rate of 0.4 mL/h for the silica precursor and 1.2 mL/h for HA, the voltage used was 15 kV and the distance between the nozzle and the collector 20 cm. For the obtainment of the ceramic fibers, the green fibers were dried at 50°C for 24 h in a stove (Thermoscientific® mod. OSG60) and later heat treated in a furnace (Thermoscientific® mod. FB1410M) at 800°C for 3 h with a temperature ramp of 0.5°C/min.

## 2.2 Characterization

The morphologies of the nanofibers were observed by a scanning electronic microscope (FE-SEM, SU5000, Hitachi). The average diameter of nanofibers was determined by analyzing the SEM images with image analyzing software Fiji [21]. Attenuated total reflectance Fourier transform infrared (ATR-FTIR) spectroscopy spectra of the samples were obtained with a spectrometer (ALPHA Platinum, Brucker Optics) in the wavenumber range 400–4000 $cm^{-1}$. X-ray diffraction (XRD) measurements were carried out to characterize the crystalline phase of $SiO_2$-HA nanofibers with a Panalytical© X'Pert Pro Alpha-1 X-ray diffractometer with Cu Kα radiation at 40 kV/30 mA. The diffractograms were scanned in a 2θ range of 10–80 at a rate of 5°/min. Surface area and pore volume of obtained samples were determined by $N_2$ physisorption and the BET equation using an equipment ASAP2010, Micromeritics. Adsorbed water was eliminated before analysis by drying at 150°C for 2 h.

## 2.3 Cell culture for *in situ* assay

"The Comite Instituiocional de Ethica y Bioetica Universidad Autonoma de Ciudad Juarez approved this study under approval number CIBE-2017-2-84." The Primary Dermal Fibroblast Normal Human, Neonatal (HDFn) was obtained from ATCC® (PCS-201-010™) and were cultured in Dulbecco's modified Eagle's medium (DMEM, SIGMA, D5523) with supplements of 10% fetal bovine serum (FBS) and 1% penicillin/streptomycin, at 37°C in a humidified atmosphere with 5% $CO_2$ and 95% air (SHELLAB, SL2406). The culture medium of both was refreshed every two days; upon confluence, cells were rinsed with 2 mL of phosphate-buffered saline (PBS) solution and incubated with 5mL of 0.05% trypsin-EDTA at 37°C in a humidified atmosphere with 5% $CO_2$. Next, within 1–2 min, the trypsin enzyme activity was stopped by the addition of 5 mL of complete growth medium and centrifuged for 5 min at 3000 rpm. The supernatant was discarded, while the cells were suspended in fresh medium and seeded onto culture flasks for further propagation and subsequent passages. Cells from 2nd to 4th passages were seeded on composites to measure its growth.

## 2.4 Cell viability assay

The $SiO_2$, HA and $SiO_2$-HA nanofibers were cut into round pieces (5.5 mm in diameter) and disinfected by exposure to UV light for 30 min on each side, then placed in a 96-well plate and seeded with 5000 cells per well; tissue culture plate was used as control. Cells viability was measured at time points of 24, 36 and 72 h using MTT reagent (3–4, 5-dimethylthiazol-2-yl) -2, 5-diphenyltetrazolium bromide, Sigma-Aldrich©), 0.5 mg/mL in PBS. [22] On the day of measurement, medium was carefully replaced on fresh DMEM + 10% FBS with diluted MTT (1:10, 10% MTT), and incubated for 1 h at 37°C in a $CO_2$ incubator to allow the transformation of MTT dye to formazan salt. After removing incubation medium, formazan crystals were dissolved in 100 μl solution of DMSO. MTT reduction was quantified by measuring the light absorbance at 570 nm using the Benchmark Plus absorbance microplate reader (Bio-Rad, Inc.). MTT test was repeated nine times. Percentage of viability was calculated by the following

formulation:

$$Viability(\%) = (OD\ Treated\ cells)/(OD\ Control\ cells) \times 100$$

### 2.5 In vivo biocompatibility test

Wistar rats with four or three-month-old male weighing approximately 300 grams were used, divided into three groups, and one rat was used as a control. Each of the rats was housed individually with the conditions established by Institutional Animal Care and Use Committee (IACUC), Comite Institucional de Etica y Bioetica, Universidad Autonoma de Ciudad Juarez (CIEB-UACJ) and the NOM-062-ZOO-1999 on technical specifications for the production, care and use of laboratory animals during the entire process of the experimental phase (Proyect: CIBE-2017-2-84). The implants were placed on the back of each animal, and the animals were sacrificed per group at 2, 4 and 6 weeks. The sample unit and analyzes were histological sections obtained from the section of the implant and from the subcutaneous cell tissue surrounding the back of the rats, the histological samples were obtained after the established times. The implants of the composite to be analyzed were prepared by taking a membrane folded into a roll of a length of 5 millimeters and 1.3 millimeters in diameter, which were sterilized for 30 minutes before implantation in UV light. Implant placement: Xylazine (PROCIN®) and ketamine (ANESKET®) were used in doses of 8 and 40 mg/Kg respectively to anesthetize the animal, the anesthesia was administered intraperitoneally, once the anesthesia took effect, a depilation of the area where the materials were implanted. The incisions to place the composite were made two centimeters apart. Once the material was implanted, the incision was closed with polyethylene suture, placing two points per incision.

Histological samples: After the established time for each study group, euthanasia was performed with an overdose of pentobarbital (Penta-Hypnol®) intraperitoneally, to take biopsies of the tissue containing the implant, which was obtained by trichromy. Once the animal's tissues were obtained, they were placed in 10% formalin (Drotasa®) to preserve and fix the tissue until its subsequent staining.

For inclusion in paraffin, the tissue was dehydrated by immersing it in ethanol solutions (Sigma-Aldrich®) at 70, 90, 96 and 100˚ for approximately 10 minutes for each solution. After dehydrating the tissue, it was placed in a solution of xylene (Sigma-Aldrich®) for 30 minutes; the tissue was transferred to liquid paraffin at approximately 60˚C with the desired orientation in to the mold, and finally the sample was allowed to solidify. The cuts of the samples were made in a microtome (KEDE®-3358) for paraffin with a thickness of $\approx$ 5 μm. The slides were fixed with a gelatin solution, once the samples were spread on the slide, drying was carried out between 35 and 40˚C for 12 h to remove the water. Hematoxylin-eosin staining was performed on the paraffin sections, a deparaffining with xylene (99%, Sigma-Aldrich®) was carried out for 10 minutes, after deparaffining it was passed to the hydration part with ethanol in a decreasing concentration from 100 to 80˚ for 10 minutes in each concentrations, once the samples were hydrated, the slide was placed in hematoxylin (MERCK®) for 3 minutes and rinsed with water, then the samples were placed in eosin (MERCK®) for 30 seconds and washed with ethanol at 80˚ for 15 seconds and next the samples were covered with the coverslip.

## 3. Results and discussion

The morphology of the fibers is shown in Fig 1. The nanofibers exhibited and interconnected pore structure. Overall, the surface of the fibers was smooth, whit only some deposits of the material amongst the fibers due to the ejection of the constituent gels. The same pattern of

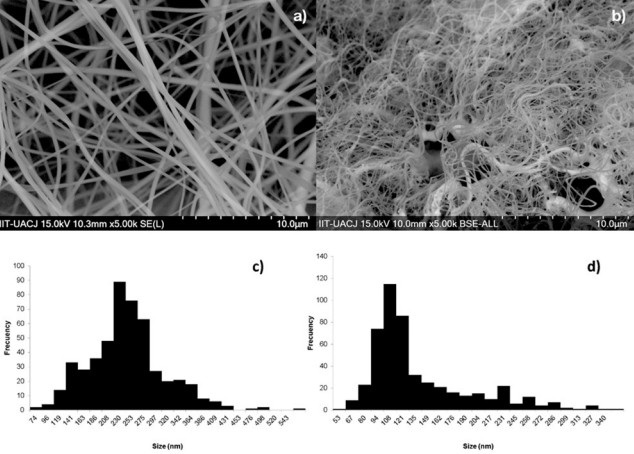

**Fig 1. SEM micrography of the SiO₂-HA composite (a) before and (b) after sintering, and respectively (c) histogram of before and after.**

contrast is observed throughout the fibers and don't show agglomerations on the surface due to the stability of the jet during electrospinning. Average diameter decreased from 230±20 nm (Fig 1a) to 110±17 nm (Fig 1b) after heat treatment at 800°C. The surface area of the sintered SiO₂-HA fibers presents a surface area of 6.57 m²/g, pore volume of 0.025 cm³/g, and pore size of 15.75 nm. According with the physisorption process the fibers have characteristic of meso-porous structure. PVP is used to aid the formation of fibers by electrospinning, and acts as a capping agent which controls nucleation and growth of hydroxyapatite crystals, for the coordination of N and O atoms in PVP structure with calcium ions.

ATR-FTIR analysis was carried out for the characterization of SiO₂ fibers, HA fibers and the SiO₂-HA fibers between 400–4000 cm⁻¹. Silica fibers spectrum in Fig 2a show characteristic bands for vibrational modes of Si-O-Si at 450, 800, 1106 and 1180 cm⁻¹. The HA fibers spectrum (Fig 2b) 550, 602 and 628 cm⁻¹ are vibrational modes to flexural vibrations assigned to $PO_4^{3-}$, bands at 961, 1022 and 1090 cm⁻¹ corresponding to symmetric phosphate group

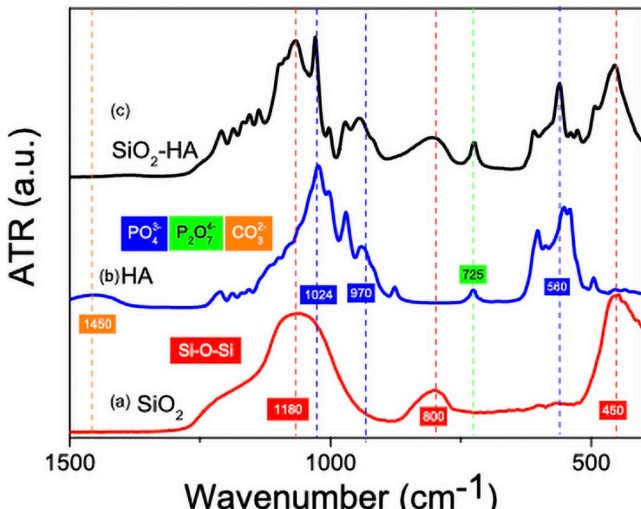

**Fig 2. IR spectrums of a) SiO₂ fibers, b) HA fibers and c) SiO₂-HA composite fibers.**

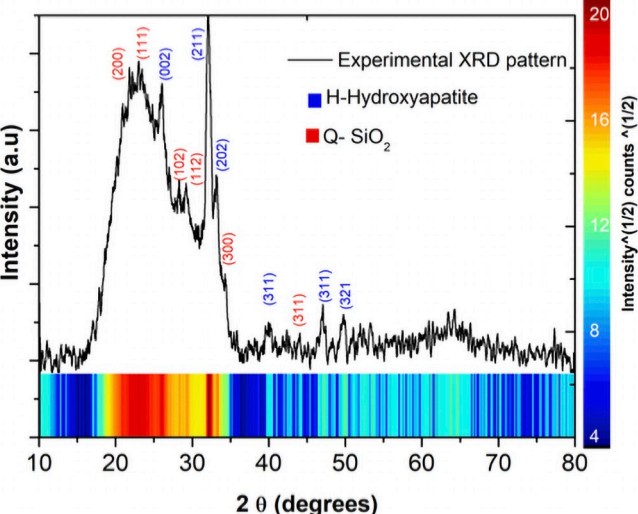

**Fig 3. XRD pattern for the SiO₂-HA composite membrane sintered at 800°C, where (H) Hydroxyapatite and (Q) Silica are noted.**

vibrations, 725 cm$^{-1}$ for $P_2O_7^4$ and 1450 cm$^{-1}$ for $CO_3^{2-}$[23]. $P_2O_7^4$ band are like pure β-tricalcium phosphate and is formed at higher temperatures, therefore it is necessary not to raise the temperature to avoid decomposition of HA [23,24]. In Fig 2c bands for vibrational modes characteristic of SiO₂ and HA show that both components form the fibers of the composite. It can also be observed that bands that phase begins to form at higher temperature, which is shown by characteristic shoulders which become more sharply and explicit.

The crystalline phases of the fibers were investigated using XRD (Fig 3). Peaks that belong to HA are located at 26.12°, 32.1°, 33.3°, 39.9°, 47.1° and 49.5°, which belong to the planes (002), (211), (202), (310), (312) and (321), respectively [24]. XRD patterns of HA are in good agreement with the standard of hydroxyapatite phase JCPDS no.00-009-0432 and confirms that HA is well crystallized. The microstructure of HA is affected by the thermal treatment since the calcination temperature increases the crystallinity. It has been reported that around 800°C the microstructure is affected and around 1000°C additional phases start to appear, such as calcium phosphate polymorphs [α, β-Ca₃(PO₄)₂] and calcium oxide (CaO), but in this case an additional phase doesn't appear because combination of amorphous silica in HA gives stability to the structure of the fibers [25]. According with IR and XRD the combination of SiO₂ an HA does not introduce a differential or partial phase transformation of HA to another tricalcium phosphate but improved the form stability due to less shrinkage after sintering of fibers in agreement with SEM results.

Fig 4a shows the theoretical XRD patters and the structural models (insert) of the SiO₂-HA and HA samples respectively. Also, the Fig 4b illustrate the experimental XRD pattern of the SiO₂-HA and HA. In the comparison between the SiO₂-HA and the HA patterns (simulated and experimental), is possible to identify some variations in the intensities at low angles diffraction. Specifically, in the SiO₂-HA XRD patterns, the intensities at 26.12° (002) increase and the Full Width at Half Maximum (FWHM) of the peaks decreases. In the experimental patterns (Fig 4a and 4b), the amorphous phase present in the SiO₂-HA decrease in the range of 15–25°. It is necessary to quantify the parameters associated to the crystallinity to establish evidence that can support the results obtained in the viability assays in relation to this property.

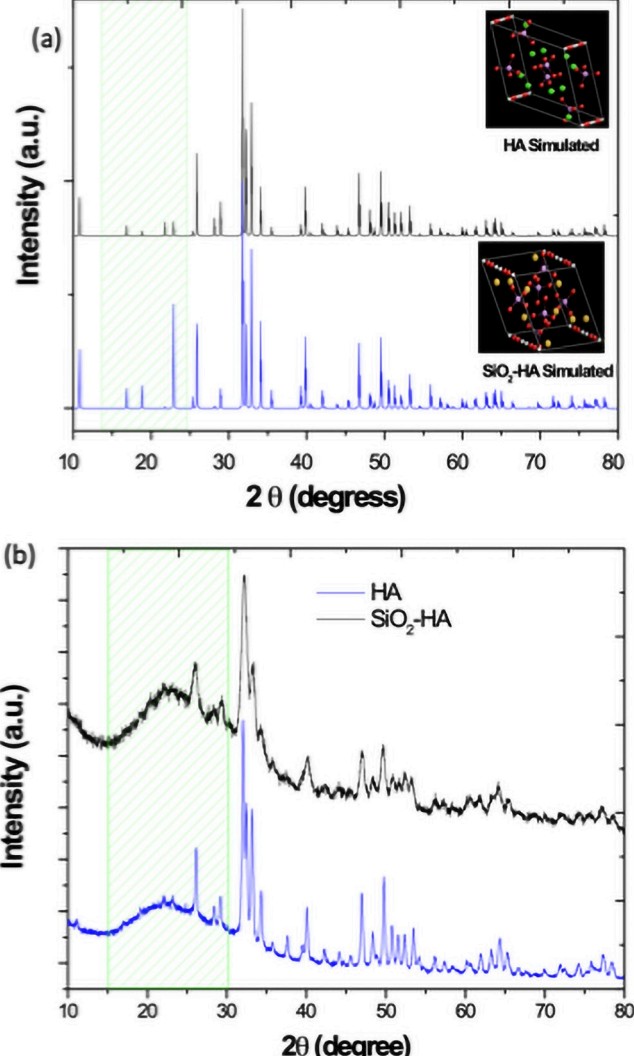

**Fig 4. a) Theoretical XRD patterns and the structural models and b) experimental XRD patterns of the HA and SiO₂-HA.**

For this reason the crystallinity percentage of the HA and SiO₂-HA experimental XRD patterns was calculated.

The crystallinity percentage (% Crystallinity) can be calculated as:

$$I_c = \frac{I_c}{I_c + I_A} * 100 \tag{1}$$

where $I_c$ is the integrated crystalline phase and $I_A$ is the integred amorphous phase.

The Pseudo-Voigt method was employed to calculate the fitted profile peaks and consequently, the area of the crystalline and amorphous phase. This method is based on the Pseudo-Voigt function, which is a convolution and the Gauss and Lorentz functions. In general form, pV function is given by [26]:

$$pV(x) = \eta G(x) + (1 - \eta)L(x) \tag{2}$$

Where G(x) and L(x) are defined as the sum of the Gaussian peak and Lorentzian peak, respectively. In this sense, is necessary to obtain the normalized peaks of Gauss ($G'(x)$) and Lorentz ($L'(x)$), the Pseudo-Voigt function can expressed as:

$$pV(x) = I[\eta G'(x, \Gamma) + (1 - \eta)L'(x)] \qquad (3)$$

Where: I is the intensity of the peak observed, $\Gamma$ is the FWHM for the Gaussian and Lorentzian peaks, $x_0$ is the peak position, $\eta$ is the Gaussian ratio.

The Lorentzian part is described by

$$L\prime(x) = \frac{1}{\pi} \frac{\Gamma/2}{(x - x_0)^2 + (\Gamma/2)^2} \qquad (4)$$

and the Gaussian part is:

$$G\prime(x) = \frac{1}{\sigma\sqrt{2\pi}} e^{-\frac{(x-x_0)^2}{2\sigma^2}} \qquad (5)$$

Substituting the Lorentzian and Gaussian part in the Pseudo-Voigt equation, is possible to obtain the expression employed during the fit profile peaks process. This expression can be written as:

$$pV(x) = h* \left[ \eta * \exp\left(-\frac{(x - x_0)^2}{-2\sigma^2}\right) + (1 - \eta)\frac{(\Gamma/2)^2}{(x - x_0)^2 + (\Gamma/2)^2} \right] \qquad (6)$$

where:

$$h = \frac{2I}{\pi\Gamma}\left[1 + (\sqrt{\pi ln2} - 1)\eta\right] \text{ and} \sigma = \frac{\Gamma}{2\sqrt{2ln2}}$$

In this sense it is possible to establish a relationship between the FWHM and the standard deviation, which can be expressed as:

$$FWHM = 2\sqrt{2ln2}\sigma$$

The Fig 5 shows graphically the FWHM values calculated from the experimental XRD patterns of the HA and SiO$_2$-HA samples. In this figure, it is possible to observe clearly that the FWHM of the SiO$_2$-HA exhibits the highest values. Consequently, the crystallinity is lower that the HA sample. Fig 5 describes also, the peak areas of the HA and SiO$_2$-HA samples, in this plot is possible to observe that crystallinity percent of the HA is 91.74% while for the SiO$_2$-HA is 84.95%. These values were obtained from the Eq 1. Therefore, the crystallinity values obtained allow us to affirm with all property that the SiO$_2$-HA sample is less crystalline than the HA sample. Thus, the cell viability of the compounds can be evaluated as a function of their crystallinity. Being the crystallinity the fundamental parameter for the discussion of the results obtained from cell viability assay presented in the subsequent part. It has been reported that the amorphous phase of SiO$_2$ exhibits a mayor biocompatibility [27] in relation to the crystalline phases.

The crystallites size of the HA and SiO$_2$-HA samples was calculated by the Scherrer equation. The crystallites size obtained from the HA and SiO$_2$-HA samples were 70 and 30 nm, respectively. In this sense, in important to mentioning that this fact is in concordance with the results observed from the crystallinity calculations. Due that the SiO$_2$-HA composite exhibits a lower crystallinity, the crystallite size also is minor due that in this sample, only the hydroxyapatite phase, determine the crystallite size.

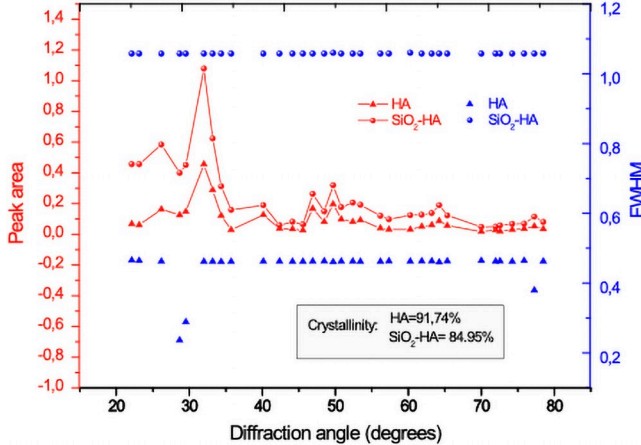

**Fig 5. Plot of the peak area and FWHM of the samples HA and SiO$_2$-HA.**

An MTT assay was carried out to prove the cytocompatibility of the thermally treated membranes. Fig 6 shows how the human neonatal fibroblasts proliferated well in a span from 24 to 72 h, having the maximum viability at 48 h when cultured in the SiO$_2$-HA membrane. While HA is capable of stimulating growth of the cells [2], the support provided by the silica allowed the anchorage of the cells [28]. It is also known that silicon acts as an interlacing agent with the ECM, allowing the fast growth and dispersion of the cells on the membrane [29]. Is reported that HA presents ions that provide the capacity of partial or complete replacement of PO$_4^{3-}$ ions by HPO$_4^{2-}$, Ca$^{2+}$ by K$^+$ or Mg$^{2+}$, and OH$^-$ by F$^-$, Cl$^-$, Br$^-$, helping in the solubility and as a source of ions which will intervene in the cellular metabolization helping the integration of the composite, and the support provided by the silica allowed the anchorage of the cells [28–30]. The SiO$_2$-HA membrane showed an in vitro behavior significantly different then the membranes of SiO$_2$ and HA as well as the control throughout the experiment. At 72 h, the viability of the cells on this material decreases, most likely because of the high increment in population, reducing the amount of space and nutrients for new cells to develop.

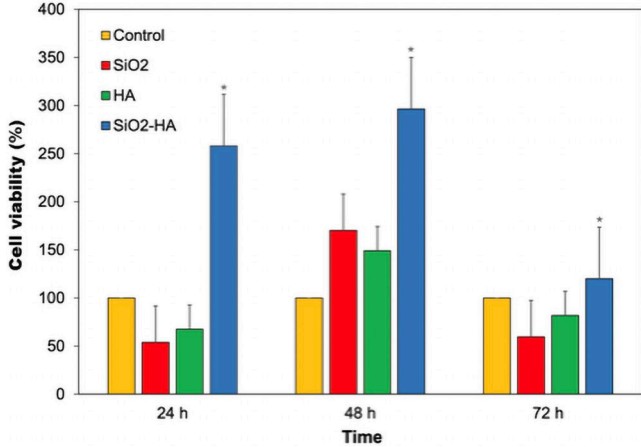

**Fig 6. Cell viability of fibroblasts incubated onto thermal-treated SiO$_2$, HA and SiO$_2$-HA membranes at 800˚C.** * means p value <0.05.

It is important to note that the highest cell viability occurs in the case of the compound $SiO_2$-HA, and whose mathematical description corresponds to a polynomial curve of the third degree, as well as the viability of hydroxyapatite. Furthermore, the cell viability of $SiO_2$ can be described using a second-degree polynomial. These results support the formation and efficiency of the $SiO_2$-HA compound, since in this compound a synergy attributable to the individual materials of $SiO_2$ and HA can be observed, even in their mathematical behavior.

Some authors have pointed out that $SiO_2$ causes a negative effect on certain cell cultures like endothelial cells and fibroblasts [31,32], most of the time as nanoparticles, which when ingested by cells can produce oxidative stress and even arrest of the cellular cycle and apoptosis [33], but this depends on size and quantity [34–37]. Since the $SiO_2$-HA material contains silica contained in hydroxyapatite while also having overall sizes higher than those reported in nanoparticles ranging below the 100 nm [31,34,38], no cytotoxicity should be observed from silica, and statistically, the behavior of the $SiO_2$ and HA membranes were similar. Ekholm *et al.* assert that materials composed by apatite type minerals can absorb proteins and signaling molecules which in this case are helpful for the overall increase in viability of the cells [39]. The high development of growth at 48 h decreases the chance for newer cells to develop, while also reducing the viability of the already present, phenomenon that can be observed at 72 h. Penttinen *et al.* suggest the combinations of hydroxyapatite and silica-based sol-gel glasses are more efficient in preparation and have a better success at a physiological level on the cells [32].

In a mixture process, we must consider components (x1, x2, . . ., xq), where the proportions of the components must sum to a unit. However, the quality of the products depends not merely on the appropriate combination of these proportions, but also on the right conditions of the c process controllable variables (w1, w2, . . ., wc). To deliver a solution to this type of problems, a polynomial function is fitted by the least square's method in a crossed array design as shown in equation [34].

$$Y = f(x, w) + \varepsilon \tag{7}$$

Controllable process variables are assumed to be continue, linear, centered and coded with mean zero. Historical or theoretical data can be employed to center them to ±1 as well as the variance, which includes the parameter estimation error of the model. The individual model terms will be tested at 5% significance level [40–44]. A mixture process variable experiment was designed in order to optimize a maximum cellular viability. HA (HA) and $SiO_2$ (SiO2) are the two mixture variables and Time are taken as process variable. HA proportions were set up at 0, 2/3, and 1 while $SiO_2$ proportion was 0, 1/3, 1. Time variable was set at 24, 48 and 72 h values. Ten replicates were measured under this condition. Table 1 gave us the regression model for bone tissue cellular grown response that indicates an interaction between HA, $SiO_2$ and

**Table 1. Regression analysis for mixture process variable model of values at 24 h and 48 h.**

| Estimated Regression Coefficients for Response (component proportions) | | | | | |
|---|---|---|---|---|---|
| Term | Coefficient | SE Coef | T | P | VIF |
| HA | 0.21814 | 0.01457 | * | * | 1.503 |
| $SiO_2$ | 0.23206 | 0.01442 | * | * | 1.117 |
| HA*$SiO_2$ | 0.80508 | 0.07954 | 10.12 | 0 | 1.59 |
| HA*time | -0.04246 | 0.01204 | -3.53 | 0.001 | 1.026 |
| | | S = 0.05944 | $R^2$ (pred) = 66.09% | $R^2$ (adj) = 69.27% | $R^2$ = 71.08% |

Time with 96.72% for $R^2_{adj}$. The model can be described by the following equation:

$$Y = 0.1536HA + 0.1572SIO2 + .9948\ HA * SIO2 + (0.0846HA + 0.0748SIO2 - 0.1654HASIO_2)\ TIME$$

The optimal response given by a maximum cellular viability percentage was located at 0.4415 units by 0.512 of HA and 0.4783 of $SiO_2$ at 48 hours of time like shows in Table 1 and Fig 7. However, if proportions varied to 0.6649 and 0.3351 for HA and $SiO_2$ respectively, the cellular viability percentage had an optimal maximum of 0.4209, which is very good.

An additional experiment under similar conditions for proportions but two levels for process variables at 48 and 72 hours was set. Table 2 shows the regression model for bone tissue cellular grown response that shows an interaction between HA, SIO2 and Time with 69.27%

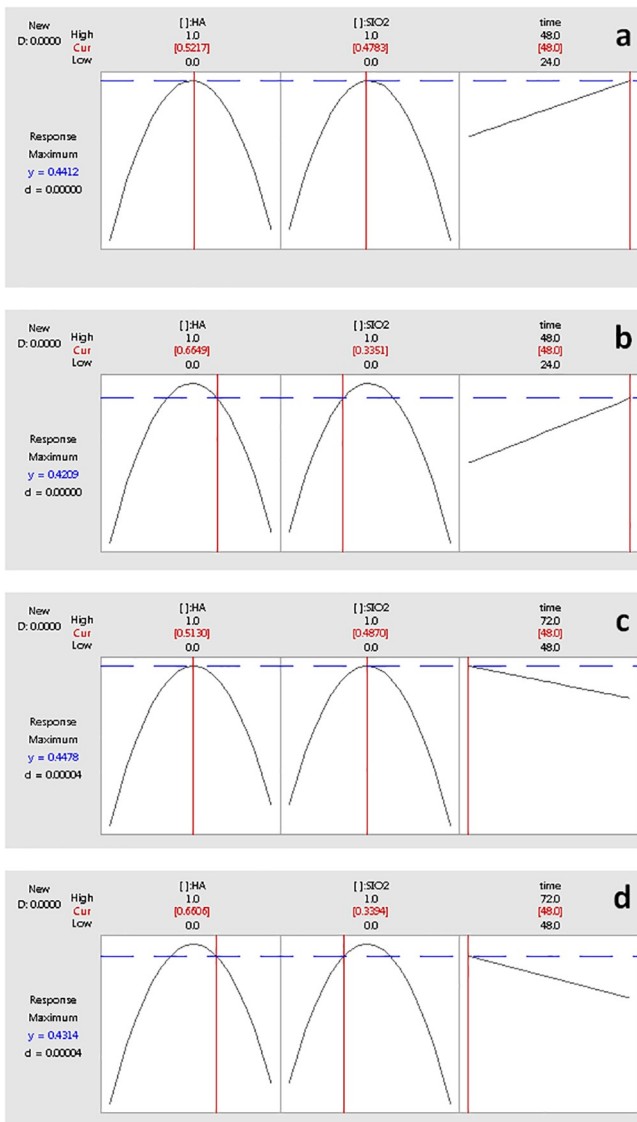

**Fig 7. Maximum Optimal Response Comparison (a) at (0.512, 0.4783, 48) versus (0.6649, 0.3351, 48), and (b) at (0.513, 0.487, 48) versus (0.6606, 0.3394, 48).**

**Table 2. Regression analysis for mixture process variable model of values at 48 h and 72 h.**

| Estimated Regression Coefficients for Response (component proportions) | | | | | |
|---|---|---|---|---|---|
| Term | Coefficient | SE Coef | T | P | VIF |
| HA | 0.1536 | 0.005596 | * | * | 1.304 |
| SiO$_2$ | 0.1572 | 0.005596 | * | * | 1.126 |
| HA*SiO$_2$ | 0.9948 | 0.040276 | 24.7 | 0 | 1.363 |
| HA*time | 0.0846 | 0.005596 | 15.12 | 0 | 1.304 |
| SiO2*time | 0.0748 | 0.005596 | 13.36 | 0 | 1.126 |
| HA*SiO$_2$*time | -0.1654 | 0.040276 | -4.11 | 0 | 1.363 |
| | | S = 0.021671 | R$^2$ (pred) = 95.7% | R$^2$ (adj) = 96.72% | R$^2$ = 97.14% |

for R$^2_{adj}$.

$$Y = 0.21814HA + 0.23206SIO2 + 0.80508\,HA * SIO2 + -0.04246HA * TIME$$

The optimal response given by a maximum cellular growth was located at 0.4478 units by 0.5130 of HA and 0.4870 of SIO2 at 48 h showed in Fig 7b. However, if proportions varied to 0.6606 and 0.3394 for HA and SIO2 respectively, the cellular growth could have produced an optimal maximum of 0.4314, which is very good. It is noted that both experiments converge at similar location points (2/3, 1/3, 48) = 0.43 units for cellular bone tissue growth.

Fig 8 shows the evolution of the implant area in Wistar rats. In Fig 8a a control rat without any trace of inflammation or injury. In Fig 8b one of the rats two weeks after implantation of

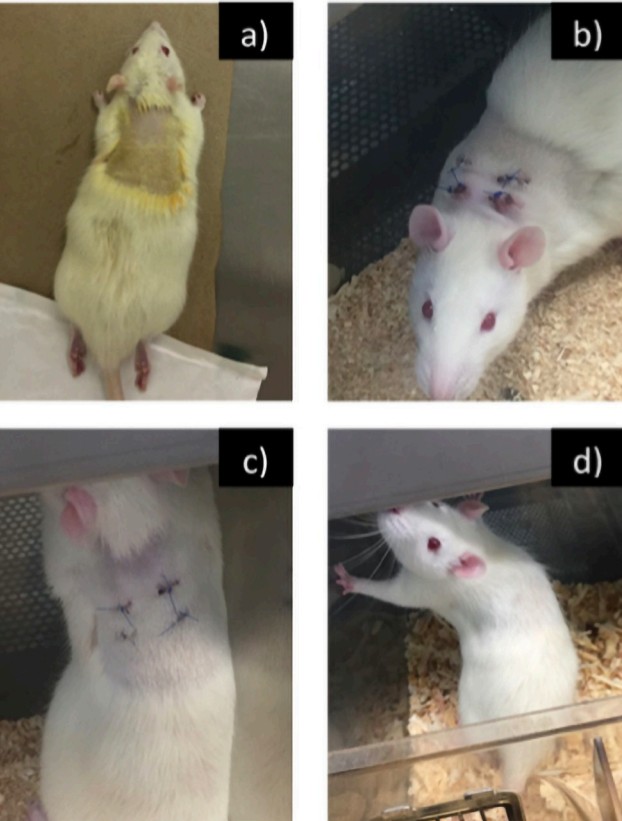

**Fig 8. a) Rat control prior to surgical intervention, b) Rat two weeks, c) Rat four weeks and d) Rat six weeks after surgery.**

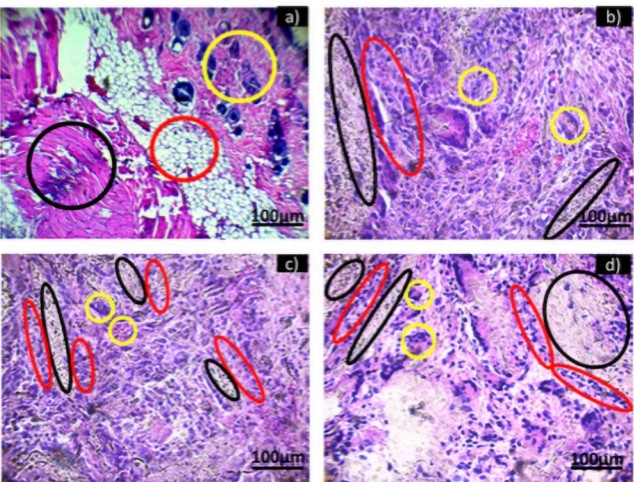

**Fig 9. Histological sections at 400 X: a) Control and exposed to HA-SiO₂ for b) two, c) four and d) six weeks.**

the material, shows inflammation in the incisions on the subcutaneous tissue, after four weeks of the surgical intervention (Fig 8c) the rat registered a significant decrease in inflammation. The already healed incisions are observed and the rat's hair have already begun to cover the lesions. Six weeks after the surgical intervention in Fig 9d the rat showed a very noticeable decrease in the inflammation in the incisions, it is noted that the rat's hair has already grown considerably, and it covers the scars. The sutures fell, indicating that the incisions have completely healed.

Fig 9 shows the control and histological sections for 2, 4 and 6 weeks of exposure to HA-SiO₂ fibers. Fig 9a shows the control tissue, which was not exposed to any material, it can be observed that the tissue does not present any type of inflammatory infiltrate, the dermis (yellow circle), the adipose tissue (red circle) and muscle tissue (black circle) can be perfectly appreciated. Fig 9b shows the tissue exposed for two weeks to HA-SiO₂, compared to the control tissue a strong chronic inflammatory infiltrate is observed, composed mainly of lymphocytes and macrophages (red circles), adjacent to the material (black circles) and multinucleated giant cells (yellow circles). Fig 10c shows the tissue exposed to HA-SiO₂ for four weeks, showing clusters of the material (black circles) and adjacent to it, a chronic inflammatory infiltrate composed of lymphocytes, macrophages (red circles) and multinucleated giant cells (yellow circles). In comparation with the two-weeks specimen, the inflammatory infiltrate has decreased. Fig 10d shows the tissue exposed to HA-SiO₂ for six weeks in which a few clumps of material (black circle) and a chronic lymphocytic inflammatory infiltrate (red circles) are present. Although the inflammatory infiltrate remains chronic, it has decreased considerably. There is no macrophage presence and the multinucleated giant cells have almost completely disappeared. After implanting a biomaterial, the body tries to heal itself causing sequentially acute inflammation, granulation, encapsulation by fibrous tissue and capsular contracture in response to the foreign body, phenomena that the analyzed tissues presented, which are considered harmful for many biomaterial applications if it does not decrease over time [45]. It was possible to observe that the tissue was still in a repair stage, therefore it is necessary to carry out histological tests at a longer exposure time to determine its biocompatibility with respect to the implantation time.

The results of this investigation should be extrapolated as promising potential biomaterial but need a more in vitro and in vivo investigations. The SiO₂-HA membranes present

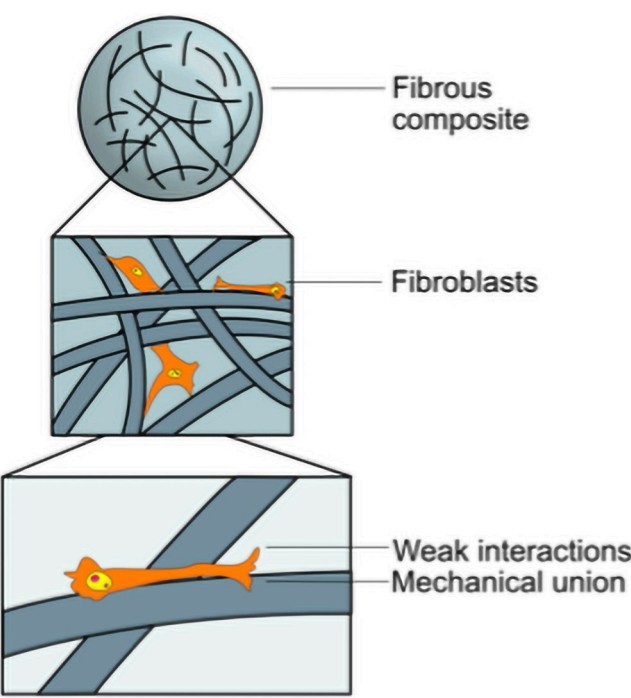

**Fig 10. Schematic presentation of various fibers surfaces with cell behavior.**

increments in viability at the incubation time growth; this acceptance at this time evaluated could be related to the HA and its capacity for cellular conduction and induction, by providing $Ca^{2+}$ ions, which help on the suspension of $HPO_4^{2-}$ allowing the assimilation of ions allowing the formation of interfacial bonds in living tissue, favoring the integration process and tissue formation and the silica allowed the anchorage of the cells.

The fibers-cell interaction is consisted two events, first is cell adhesion and spreading, and second events are related to cell proliferation. Cell adhesion is important for the interaction, in Fig 10 a schematic presentation of fibers surface with cell behavior is represented. Cell adhesions give the process for the interaction and bind to a material surface for another cells and is necessary for cell communication to organ formation or tissue maintenance. The advantage of electrospinning is the ability to produce nanofiber scaffolds or oriented fibers by changing the manufacturing parameters. Oriented electrospun fibers can induce cell orientation, which is why the development of new ceramic scaffolds is of great interest in biomedical engineering.

## Conclusion

In the last years, an effort has been made to produce materials which aid in the recovery of damaged tissue. Progress in the creation of bioceramics is the good combination of properties of their raw materials. The fibers obtained in this study are effective as an alternative for bioceramics scaffolds manufactured by the sol gel and electrospinning technique. Silica and hydroxyapatite combination improve the bioactivity significantly different then the membranes of $SiO_2$ and HA. The precise fibrillar and porous design demonstrated the advantages for the combination the sol gel and electrospinning techniques for promote a high percentage of viability on a fibroblast cell line.

## Supporting information

**S1 Fig. Graphical abstract.**
(TIFF)

**S1 Data. Table 2 shows the calculated values of FWHM, area of the crystalline phase and the complementary parameter for the XRD peak fitting.**
(DOCX)

## Acknowledgments

PRODEP, Universidad Autónoma de Ciudad Juárez and CONACYT. Reyes-Lopez appreciates the assistance of Dr. Alejandro Donohue Cornejo and Dr. Juan Carlos Cuevas Gonzalez for the histopathological studies.

## Author Contributions

**Conceptualization:** Jesús Alberto Garibay-Alvarado, Simón Yobanny Reyes-López.

**Formal analysis:** Ericka Berenice Herrera-Ríos, Claudia Lucía Vargas-Requena, Simón Yobanny Reyes-López.

**Funding acquisition:** Simón Yobanny Reyes-López.

**Investigation:** Jesús Alberto Garibay-Alvarado, Simón Yobanny Reyes-López.

**Methodology:** Jesús Alberto Garibay-Alvarado, Ericka Berenice Herrera-Ríos, Claudia Lucía Vargas-Requena, Álvaro de Jesús Ruíz-Baltazar, Simón Yobanny Reyes-López.

**Project administration:** Simón Yobanny Reyes-López.

**Resources:** Simón Yobanny Reyes-López.

**Supervision:** Álvaro de Jesús Ruíz-Baltazar, Simón Yobanny Reyes-López.

**Writing – original draft:** Jesús Alberto Garibay-Alvarado, Simón Yobanny Reyes-López.

**Writing – review & editing:** Jesús Alberto Garibay-Alvarado, Álvaro de Jesús Ruíz-Baltazar, Simón Yobanny Reyes-López.

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
