## [Decision Letter · Decision Letter 0]

3 Aug 2020

PONE-D-20-16200

Bioactive Silica-Hydroxyapatite coaxial composite scaffold for bone tissue engineering

PLOS ONE

Dear Dr. Reyes-López,

Thank you for submitting your manuscript to PLOS ONE. After careful consideration, we feel that it has merit but does not fully meet PLOS ONE’s publication criteria as it currently stands. Therefore, we invite you to submit a revised version of the manuscript that addresses the points raised during the review process.

It has been reviewed by experts in the field and we request that you make Major Revision before it is processed further. Please find your manuscript and the review reports below:

**Reviewer 1**

Some typing errors and grammatical errors present throughout the manuscript. Methodology for in vivo study does not clearly stated the sample size. Objectives not clearly stated both in the abstract and introduction, especially not mention anything about characterization of the material and in vivo study.

**Reviewer 2**

In this research, investigate the cytotoxic behavior of electrospun SiO2-HA non-woven ceramic membranes. However, some issues regarding present study should be addressed before publication in PLOS ONE. It is recommended that the following be done to complete the article:

1-“Abstract” should be rewritten again by considering better grammar.

2- Generally, using chemical formulas such as ‘SiO2” is not usual in “Keywords”. Therefore, another keyword should be replaced instead of “SiO2”.

3- What does the surface area of SiO2-HA fiber indicate? Please discuss the issue?

4- Results of ATR-FTIR should be mentioned with references. Please examine the issue.

5- Statistical analysis or p-value is required for in vitro and in vivo tests. Please mention the issue in the manuscript.

6- Results of in vitro and in vivo should be expressed in “Conclusion”. “Conclusion” should be rewritten again.

7- To conclude, writing related to manuscript is weak. Please examine and use better grammar for it.

**Reviewer & Editor Comments**

Q1: In the abstract, you need to focus more on the quantitative information, not qualitative ones.

Q2: In the “Introduction” section, the authors were supposed to clear the answers of the following questions;

a. Which exact problem was supposed to be solved by the present research?

b. Which new achievement(s) was supposed to be obtained by the present research in comparison with the previous reports?

Q3: Title and subtitle of each section require numbering.

Q4:  Please describe the underlying reason behind selection of silica precursor at 10 w/v% and HA sol at 20 w/v% for the composite fibers.

Q5: You stated that " The surface area of the sintered SiO2-HA fibers was 5.77 m2/g, with an

average pore width of 135.87 Å". How did you measure surface area and averages pore width

of the sintered SiO2-HA fibers.

Q6: In the method section, how many times have you repeated the cell culture tests? Have you considered the repeatability of the tests?

Q7: Histogram analysis of SiO2-HA fiber composite to further confirm the average diameter of fiber composite and its distribution.

Q8: What do you mean by "vibrational modes assigned to "...725 cm-1 for P2O74-". There is no such band "P2O74-" in aforesaid wavenumber.

Q9: "vibrational modes of Si-O-Si" should be indicated in the ATR-FTIR analysis". In addition, your statement is not clear "according with IR and XRD the combination of SiO2 an HA does not introduce a differential or partial phase transformation of HA to another tricalcium phosphate".

Q10: I recommended the author to present the value of "crystallinity percentage", and "crystalite size" of HA and SiO2-HA in the table instead of Fig. 5.

Q11: the author should check the value of cell viability of fibroblasts incubated onto thermal-treated SiO2, HA and SiO2-HA. You present high cell viability of fibroblasts regarding SiO2, HA and SiO2-HA, these values should be compared with other study and literature. It is worth noting that you stated that  "SiO2 has a negative effect on certain cell cultures such as endothelial and fibroblast [29, 30]" which has a contradiction with  presented value in the graph (Fig. 6).

Q12: Y axis of Fig. 6 is missing. In addition, "*" sign is missing too inside the graph.

Q13: Table 1 and Table 2 regarding "Regression Analysis for Mixture Process Variable Model of values at 24 h and 48 h" could be presented in the supporting information.

Q14: Fig. 7 and Fig. 8 regarding maximum optimal response should merge together and be presented as Fig. 7a, b.

Q15: For benefit of readers, the schematic illustration regarding schematic demonstration of the interactions between the fibroblasts cell and surface of SiO2-HA fiber composite could be presented.

Q16: In conclusion part, you need to present more quantitative data in order to make it easier and convenient for readers to compare the specimens.

Q17: Moreover, for possible final paper acceptance, it is requested that authors improve the English

presentation, both grammar and style. Authors are strongly advised to have their paper revised for language by a native English or equivalent expert.

Q18: Surprisingly small reference to PLOS ONE in the literature despite the large relevant literature there. This should be improved. There are several important papers in the recent literature

" between 400-4000 cm-1" should change to " between 400-4000 cm-1"

" cells. [27,28]" should change to " cells [27,28]."

"membrane. [28]" should change to "membrane [28]."  

We look forward to receiving your revised manuscript.

Kind regards,

Hamid Reza Bakhsheshi-Rad

Academic Editor

PLOS ONE

Journal Requirements:

"no"

Reviewers' comments:

Reviewer's Responses to Questions

**Comments to the Author**

1. Is the manuscript technically sound, and do the data support the conclusions?

Reviewer #1: Partly

Reviewer #2: Yes

2. Has the statistical analysis been performed appropriately and rigorously? 

Reviewer #1: No

Reviewer #2: No

3. Have the authors made all data underlying the findings in their manuscript fully available?

Reviewer #1: No

Reviewer #2: Yes

4. Is the manuscript presented in an intelligible fashion and written in standard English?

Reviewer #1: No

Reviewer #2: No

5. Review Comments to the Author

Reviewer #1: Some typing errors and grammatical errors present throughout the manuscript.

Methodology for in vivo study does not clearly stated the sample size.

Objectives not clearly stated both in the abstract and introduction, especially not mention anything about charaterization of the material and in vivo study.

Reviewer #2: Publish after major revision

In this research, investigate the cytotoxic behavior of electrospun SiO2-HA non-woven ceramic membranes. However, some issue regarding present study should be address before publication in PLOS ONE. It is recommended that the following be done to complete the article:

1-“Abstract” should be rewritten again by considering better grammar.

2- Generally, using chemical formula such as ‘SiO2” is not usual in “Keywords”. Therefore, another keyword should be replaced instead of “SiO2”.

3- What does the surface area of SiO2-HA fiber indicate? Please discuss about the issue?

4- Results of ATR-FTIR should be mentioned with references. Please examine the issue.

5- Statistical analysis or p-value is required for in vitro and in vivo tests. Please mention the issue in the manuscript.

6- Results of in vitro and in vivo should be expressed in “Conclusion”. “Conclusion” should be rewritten again.

7- To conclude, writing related to manuscript is weak. Please examine and use better grammar for it.

6. PLOS authors have the option to publish the peer review history of their article (what does this mean?). If published, this will include your full peer review and any attached files.

Reviewer #1: No

Reviewer #2: No

---

## [Author Response · Author response to Decision Letter 0]

28 Aug 2020

Response comments

Reviewer 1

Some typing errors and grammatical errors present throughout the manuscript. Methodology for in vivo study does not clearly stated the sample size. Objectives not clearly stated both in the abstract and introduction, especially not mention anything about characterization of the material and in vivo study.

Response: The corrections were made in the manuscript

Reviewer 2

In this research, investigate the cytotoxic behavior of electrospun SiO2-HA non-woven ceramic membranes. However, some issues regarding present study should be addressed before publication in PLOS ONE. It is recommended that the following be done to complete the article:

1-“Abstract” should be rewritten again by considering better grammar.

Response: The corrections were made in the manuscript

2- Generally, using chemical formulas such as ‘SiO2” is not usual in “Keywords”. Therefore, another keyword should be replaced instead of “SiO2”.

Response: The corrections were made in the manuscript

3- What does the surface area of SiO2-HA fiber indicate? Please discuss the issue?

Response: The corrections were made in the manuscript

4- Results of ATR-FTIR should be mentioned with references. Please examine the issue.

Response: The corrections were made in the manuscript

5- Statistical analysis or p-value is required for in vitro and in vivo tests. Please mention the issue in the manuscript.

Response: The corrections were made in the manuscript

6- Results of in vitro and in vivo should be expressed in “Conclusion”. “Conclusion” should be rewritten again.

Response: The corrections were made in the manuscript

7- To conclude, writing related to manuscript is weak. Please examine and use better grammar for it.

Response: The corrections were made in the manuscript

Reviewer & Editor Comments

Q1: In the abstract, you need to focus more on the quantitative information, not qualitative ones.

Response: The corrections were made in the manuscript

Q2: In the “Introduction” section, the authors were supposed to clear the answers of the following questions;

a. Which exact problem was supposed to be solved by the present research?

b. Which new achievement(s) was supposed to be obtained by the present research in comparison with the previous reports?

Response: The corrections were made in the manuscript

Q3: Title and subtitle of each section require numbering.

Response: The corrections were made in the manuscript

Q4: Please describe the underlying reason behind selection of silica precursor at 10 w/v% and HA sol at 20 w/v% for the composite fibers.

Response: The corrections were made in the manuscript

Q5: You stated that " The surface area of the sintered SiO2-HA fibers was 5.77 m2/g, with an average pore width of 135.87 Å". How did you measure surface area and averages pore width of the sintered SiO2-HA fibers.

Response: The corrections were made in the manuscript

Q6: In the method section, how many times have you repeated the cell culture tests? Have you considered the repeatability of the tests?

Response: The corrections were made in the manuscript

Q7: Histogram analysis of SiO2-HA fiber composite to further confirm the average diameter of fiber composite and its distribution.

Response: The corrections were made in the manuscript

Q8: What do you mean by "vibrational modes assigned to "...725 cm-1 for P2O74-". There is no such band "P2O74-" in aforesaid wavenumber.

Response: The corrections were made in the manuscript

Q9: "vibrational modes of Si-O-Si" should be indicated in the ATR-FTIR analysis". In addition, your statement is not clear "according with IR and XRD the combination of SiO2 an HA does not introduce a differential or partial phase transformation of HA to another tricalcium phosphate".

Response: The corrections were made in the manuscript

Q10: I recommended the author to present the value of "crystallinity percentage", and "crystalite size" of HA and SiO2-HA in the table instead of Fig. 5.

Response: The corrections were made in the manuscript

Q11: the author should check the value of cell viability of fibroblasts incubated onto thermal-treated SiO2, HA and SiO2-HA. You present high cell viability of fibroblasts regarding SiO2, HA and SiO2-HA, these values should be compared with other study and literature. It is worth noting that you stated that "SiO2 has a negative effect on certain cell cultures such as endothelial and fibroblast [29, 30]" which has a contradiction with presented value in the graph (Fig. 6).

Response: The corrections were made in the manuscript

Q12: Y axis of Fig. 6 is missing. In addition, "*" sign is missing too inside the graph.

Response: The corrections were made in the manuscript

Q13: Table 1 and Table 2 regarding "Regression Analysis for Mixture Process Variable Model of values at 24 h and 48 h" could be presented in the supporting information.

Response: The corrections were made in the manuscript

Q14: Fig. 7 and Fig. 8 regarding maximum optimal response should merge together and be presented as Fig. 7a, b.

Response: The corrections were made in the manuscript

Q15: For benefit of readers, the schematic illustration regarding schematic demonstration of the interactions between the fibroblasts cell and surface of SiO2-HA fiber composite could be presented.

Q16: In conclusion part, you need to present more quantitative data in order to make it easier and convenient for readers to compare the specimens.

Response: The corrections were made in the manuscript

Q17: Moreover, for possible final paper acceptance, it is requested that authors improve the English

presentation, both grammar and style. Authors are strongly advised to have their paper revised for language by a native English or equivalent expert.

Response: The corrections were made in the manuscript

Q18: Surprisingly small reference to PLOS ONE in the literature despite the large relevant literature there. This should be improved. There are several important papers in the recent literature

" between 400-4000 cm-1" should change to " between 400-4000 cm-1"

" cells. [27,28]" should change to " cells [27,28]."

"membrane. [28]" should change to "membrane [28]."

Response: The corrections were made in the manuscript

---

## [Editor Report · Decision Letter 1]

18 Sep 2020

PONE-D-20-16200R1

Bioactive Silica-Hydroxyapatite coaxial composite scaffold for bone tissue engineering

PLOS ONE

Dear Dr. Reyes-López,

Thank you for submitting your manuscript to PLOS ONE. After careful consideration, we feel that it has merit but does not fully meet PLOS ONE’s publication criteria as it currently stands. Therefore, we invite you to submit a revised version of the manuscript that addresses the points raised during the review process.

However, for sending the decision letter the authors should revise the manuscript to address the comments received from the reviewer, point-by-point and sufficiently described below each comment and then highlight the revised section in the manuscript. All the changes should mark in **different colors** in the revised manuscript. The response to the reviewers in this form is not acceptable and should be modified according to the aforementioned comments and resubmit the revised manuscript.

We look forward to receiving your revised manuscript.

Kind regards,

Hamid Reza Bakhsheshi-Rad

Academic Editor

PLOS ONE

---

## [Author Response · Author response to Decision Letter 1]

27 Sep 2020

Response comments

Reviewer 1

Some typing errors and grammatical errors present throughout the manuscript. Methodology for in vivo study does not clearly stated the sample size. Objectives not clearly stated both in the abstract and introduction, especially not mention anything about characterization of the material and in vivo study.

Response: The corrections were made in the manuscript

Reviewer 2

In this research, investigate the cytotoxic behavior of electrospun SiO2-HA non-woven ceramic membranes. However, some issues regarding present study should be addressed before publication in PLOS ONE. It is recommended that the following be done to complete the article:

1-“Abstract” should be rewritten again by considering better grammar.

Response: The corrections were made in the manuscript

2- Generally, using chemical formulas such as ‘SiO2” is not usual in “Keywords”. Therefore, another keyword should be replaced instead of “SiO2”.

Response: The corrections were made in the manuscript

3- What does the surface area of SiO2-HA fiber indicate? Please discuss the issue?

Response: The corrections were made in the manuscript

4- Results of ATR-FTIR should be mentioned with references. Please examine the issue.

Response: The corrections were made in the manuscript

5- Statistical analysis or p-value is required for in vitro and in vivo tests. Please mention the issue in the manuscript.

Response: The corrections were made in the manuscript

6- Results of in vitro and in vivo should be expressed in “Conclusion”. “Conclusion” should be rewritten again.

Response: The corrections were made in the manuscript

7- To conclude, writing related to manuscript is weak. Please examine and use better grammar for it.

Response: The corrections were made in the manuscript

Reviewer & Editor Comments

Q1: In the abstract, you need to focus more on the quantitative information, not qualitative ones.

Response: The corrections were made in the manuscript

Q2: In the “Introduction” section, the authors were supposed to clear the answers of the following questions;

a. Which exact problem was supposed to be solved by the present research?

b. Which new achievement(s) was supposed to be obtained by the present research in comparison with the previous reports?

Response: The corrections were made in the manuscript

Q3: Title and subtitle of each section require numbering.

Response: The corrections were made in the manuscript

Q4: Please describe the underlying reason behind selection of silica precursor at 10 w/v% and HA sol at 20 w/v% for the composite fibers.

Response: The corrections were made in the manuscript

Q5: You stated that " The surface area of the sintered SiO2-HA fibers was 5.77 m2/g, with an average pore width of 135.87 Å". How did you measure surface area and averages pore width of the sintered SiO2-HA fibers.

Response: The corrections were made in the manuscript

Q6: In the method section, how many times have you repeated the cell culture tests? Have you considered the repeatability of the tests?

Response: The corrections were made in the manuscript

Q7: Histogram analysis of SiO2-HA fiber composite to further confirm the average diameter of fiber composite and its distribution.

Response: The corrections were made in the manuscript

Q8: What do you mean by "vibrational modes assigned to "...725 cm-1 for P2O74-". There is no such band "P2O74-" in aforesaid wavenumber.

Response: The corrections were made in the manuscript

Q9: "vibrational modes of Si-O-Si" should be indicated in the ATR-FTIR analysis". In addition, your statement is not clear "according with IR and XRD the combination of SiO2 an HA does not introduce a differential or partial phase transformation of HA to another tricalcium phosphate".

Response: The corrections were made in the manuscript

Q10: I recommended the author to present the value of "crystallinity percentage", and "crystalite size" of HA and SiO2-HA in the table instead of Fig. 5.

Response: The corrections were made in the manuscript

Q11: the author should check the value of cell viability of fibroblasts incubated onto thermal-treated SiO2, HA and SiO2-HA. You present high cell viability of fibroblasts regarding SiO2, HA and SiO2-HA, these values should be compared with other study and literature. It is worth noting that you stated that "SiO2 has a negative effect on certain cell cultures such as endothelial and fibroblast [29, 30]" which has a contradiction with presented value in the graph (Fig. 6).

Response: The corrections were made in the manuscript

Q12: Y axis of Fig. 6 is missing. In addition, "*" sign is missing too inside the graph.

Response: The corrections were made in the manuscript

Q13: Table 1 and Table 2 regarding "Regression Analysis for Mixture Process Variable Model of values at 24 h and 48 h" could be presented in the supporting information.

Response: The corrections were made in the manuscript

Q14: Fig. 7 and Fig. 8 regarding maximum optimal response should merge together and be presented as Fig. 7a, b.

Response: The corrections were made in the manuscript

Q15: For benefit of readers, the schematic illustration regarding schematic demonstration of the interactions between the fibroblasts cell and surface of SiO2-HA fiber composite could be presented.

Q16: In conclusion part, you need to present more quantitative data in order to make it easier and convenient for readers to compare the specimens.

Response: The corrections were made in the manuscript

Q17: Moreover, for possible final paper acceptance, it is requested that authors improve the English

presentation, both grammar and style. Authors are strongly advised to have their paper revised for language by a native English or equivalent expert.

Response: The corrections were made in the manuscript

Q18: Surprisingly small reference to PLOS ONE in the literature despite the large relevant literature there. This should be improved. There are several important papers in the recent literature

" between 400-4000 cm-1" should change to " between 400-4000 cm-1"

" cells. [27,28]" should change to " cells [27,28]."

"membrane. [28]" should change to "membrane [28]."

Response: The corrections were made in the manuscript

---

## [Editor Report · Decision Letter 2]

27 Oct 2020

PONE-D-20-16200R2

Bioactive Silica-Hydroxyapatite coaxial composite scaffold for bone tissue engineering

PLOS ONE

Dear Dr. Reyes-López,

Thank you for submitting your manuscript to PLOS ONE. After careful consideration, we feel that it has merit but does not fully meet PLOS ONE’s publication criteria as it currently stands. Therefore, we invite you to submit a revised version of the manuscript that addresses the points raised during the review process.

Specific questions and comments are listed as follow:

Q1: As mentioned previously, you stated that "The surface area of the sintered SiO2-HA fibers was 5.77 m2/g, with an average pore width of 135.87 Å". How did you measure the surface area and averages pore width of the sintered SiO2-HA fibers? The procedure should be presented in the materials and methods.

Q2: As mentioned previously, the histogram analysis of SiO2-HA fiber composite to further confirm the average diameter of fiber composite and its distribution. The histogram analysis of SiO2-HA fiber composite was not provided.

Q3: You stated that "vibrational modes assigned to "...725 cm-1 for P2O74-"relate to the HA fibers spectrum. However, this vibration mode "P2O74-" is related to FTIR spectra of commercial product (Fluka β-TCP) according to the Ref [23].

Q4: As mentioned previously, I recommended the author to present the value of "crystallinity percentage", and "crystallite size" of HA and SiO2-HA in the table instead of Fig. 5. However, the crystallite size of HA and SiO2-HA was not presented.

Q5: "PO43- ions by HPO42-, Ca2+ by K+ or Mg2+, and OH- by F-, Cl-, Br-". Upper-case and lower-case should be used to indicate ions.

Q6: For readers' benefit, the schematic illustration regarding schematic demonstration of the interactions between the fibroblasts cell and surface of SiO2-HA could be presented.

We look forward to receiving your revised manuscript.

Kind regards,

Hamid Reza Bakhsheshi-Rad

Academic Editor

PLOS ONE

---

## [Author Response · Author response to Decision Letter 2]

3 Jan 2021

The files are updated with corrections 29 Dic 2020

we send clean manuscript

1. Please amend the title either on the online submission form or in your manuscript so that they are identical.

response:

the name of manuscript is updated

2. Please upload a Response to Reviewers letter which should include a point by point response to each of the points made by the Editor and / or Reviewers. (This should be uploaded as a 'Response to Reviewers' file type.) Please follow this link for more information: http://blogs.PLOS.org/everyone/2011/05/10/how-to-submit-your-revised-manuscript/

3. Thank you so much for providing information on your IACUC. At this time, can you please update your Materials and Methods section to include that your study was specifically approved by COMITE INSTITUCIONAL DE ETICA Y BIOETICA, UNIVERSIDAD AUTONOMA DE CIUDAD JUAREZ?

Response:

the manuscript is updated

Response for specific questions and comments for Oct 27 2020 are listed as follow:

Q1: As mentioned previously, you stated that "The surface area of the sintered SiO2-HA fibers was 5.77 m2/g, with an average pore width of 135.87 Å". How did you measure the surface area and averages pore width of the sintered SiO2-HA fibers? The procedure should be presented in the materials and methods.

Response: The corrections was made in the manuscript, In the revised manuscript, the text was left in red with the mention of the technique used.

Q2: As mentioned previously, the histogram analysis of SiO2-HA fiber composite to further confirm the average diameter of fiber composite and its distribution. The histogram analysis of SiO2-HA fiber composite was not provided.

Response: The corrections was made in the manuscript, In the revised manuscript, histogram analysis of SiO2-HA fiber composite was provided

Q3: You stated that "vibrational modes assigned to "...725 cm-1 for P2O74-"relate to the HA fibers spectrum. However, this vibration mode "P2O74-" is related to FTIR spectra of commercial product (Fluka β-TCP) according to the Ref [23].

Response: The corrections was made in the manuscript, In the revised manuscript, the text in red have a new information.

Q4: As mentioned previously, I recommended the author to present the value of "crystallinity percentage", and "crystallite size" of HA and SiO2-HA in the table instead of Fig. 5. However, the crystallite size of HA and SiO2-HA was not presented.

Response: The corrections was made in the manuscript, In the revised manuscript, the text in red have a new information.

Q5: "PO43- ions by HPO42-, Ca2+ by K+ or Mg2+, and OH- by F-, Cl-, Br-". Upper-case and lower-case should be used to indicate ions.

Response: The corrections was made in the manuscript,

Q6: For readers' benefit, the schematic illustration regarding schematic demonstration of the interactions between the fibroblasts cell and surface of SiO2-HA could be presented.

Response: The corrections was made in the manuscript, an schematic demonstration of the interactions between the fibroblasts cell and surface of SiO2-HA was presented.

---

## [Editor Report · Decision Letter 3]

19 Jan 2021

Cell behavior on Silica-Hydroxyapatite coaxial composite

PONE-D-20-16200R3

Dear Dr. Reyes-López,

We’re pleased to inform you that your manuscript has been judged scientifically suitable for publication and will be formally accepted for publication once it meets all outstanding technical requirements.

Kind regards,

Hamid Reza Bakhsheshi-Rad

Academic Editor

PLOS ONE
---

## [Editor Report · Acceptance letter]

26 Jan 2021

PONE-D-20-16200R3 

Cell behavior on Silica-Hydroxyapatite coaxial composite 

Dear Dr. Reyes-López:

I'm pleased to inform you that your manuscript has been deemed suitable for publication in PLOS ONE. Congratulations! Your manuscript is now with our production department. 

Kind regards, 

on behalf of

Dr. Hamid Reza Bakhsheshi-Rad 

Academic Editor

PLOS ONE